# From Sensors to Care: How Robotic Skin Is Transforming Modern Healthcare—A Mini Review

**DOI:** 10.3390/s25092895

**Published:** 2025-05-03

**Authors:** Yuting Zhu, Wendy Moyle, Min Hong, Kean Aw

**Affiliations:** 1School of Engineering, University of Southern Queensland, Springfield, QLD 4300, Australia; min.hong@unisq.edu.au; 2School of Nursing and Midwifery, Griffith University, Nathan, QLD 4111, Australia; w.moyle@griffith.edu.au; 3Department of Mechanical and Mechatronics Engineering, University of Auckland, Auckland 1010, New Zealand; k.aw@auckland.ac.nz

**Keywords:** robotic skin, healthcare, rehabilitation, companion robots, healthcare robots, tactile sensing

## Abstract

In recent years, robotics has made notable progress, becoming an essential component of daily life by facilitating complex tasks and enhancing human experiences. While most robots have traditionally featured hard surfaces, the growing demand for more comfortable and safer human–robot interactions has driven the development of soft robots. One type of soft robot, which incorporates innovative skin materials, transforms rigid structures into more pliable and adaptive forms, making them better suited for interacting with humans. Especially in healthcare and rehabilitation, robotic skin technology has gained substantial attention, offering transformative solutions for improving the functionality of prosthetics, exoskeletons, and companion robots. Although replicating the complex sensory functions of human skin remains a challenge, ongoing research in soft robotics focuses on developing sensors that mimic the softness and tactile sensitivity necessary for effective interaction. This review provides a narrative analysis of current trends in robotic skin development, specifically tailored for healthcare and rehabilitation applications, including skin types of sensor technologies, materials, challenges, and future research directions in this rapidly developing field.

## 1. Introduction

For years, robots have featured in movies with interpretations of a faraway future, such as the Star Wars or Terminator movies, where robots have evolved as increasingly intelligent and advanced. From the first droids of Star Wars to the most recent ones, they all have one primary purpose: to assist and help humans. Many different types are shown, from droids built to serve, medical robots, and even engineering robots.

Although there are no robots currently available that are as sophisticated as those featured in the Star Wars or Terminator movies, robots are becoming more practical, and many are used in our everyday lives. Over the past three decades, robots have been integrated with various sensors that can sense force, contact, temperature, etc. However, many functions still need to be developed to mimic human skills, particularly those required to replicate human abilities, such as handling delicate or fragile objects and ensuring safe interaction with vulnerable groups like children and the elderly [1,2]. This paper overviews some of the soft robotic skin developments and outlines the key challenges that need to be overcome to advance robotic capabilities.

Robotics has made significant advancements over the past year, becoming an integral part of our daily lives by assisting with complex tasks and enhancing our experiences [3,4]. While most robots traditionally feature hard surfaces, there has been a growing interest in soft robots due to their comfort and safety features [5,6]. These soft robots use “skin” materials that transform rigid, hard-surface robots into more pliable and adaptive structures [7,8,9]. Consequently, robotic skin technology has become increasingly significant in healthcare and rehabilitation, offering innovative solutions for enhancing prosthetics, exoskeletons, and companion robots [10,11]. Many robotic applications have been developed and used in healthcare, biomedical, and companion robots. For example, humanoid robots can act in healthcare as companions, helping people with different tasks. Most existing robots have rigid surfaces/frames and are complex and expensive. One solution for robots that react to humans’ everyday living is that they should be soft and human-like [12], capable of the sense of touch. Sensors need to be soft, robust, lightweight, and inexpensive for robotic applications. Robots are featured with sophisticated technology, such as facial expression, voice recognition, self-learning, body manipulation, etc. However, current robots lack a human-like sense of touch, and their lack of softness has always limited their interaction with humans. The demand for healthcare robots has increased in recent years, particularly in aged care facilities. These robotic systems assist with rehabilitation, older adult care, and surgical procedures, offering precision, reliability, and a reduced workload for healthcare professionals. However, our research indicates that robots often lack the tactile perception for safe and effective human interactions [13,14,15]. Most commercially available humanoid robots also feature metallic or plastic exteriors with limited sensing capabilities.

This paper provides a narrative review of the current trends in robotic skin development over the past three decades, beginning in the 1990s, specifically for healthcare and rehabilitation applications, including its role in companion and healthcare robots. This review focuses on examining sensor-based robotic skin types while highlighting the key materials, technologies, challenges, and future research directions in this rapidly growing field.

## 2. Types of Robotic Skin in Healthcare and Rehabilitation

Current research in soft robotics is focused on developing sensors that replicate the softness and tactile sensitivity of human skin so that it is safer to interact with humans [10]. As an essential part of our body, human skin detects a variety of sensations, including temperature, pressure, and texture, enabling us to interact with and respond to our environment effectively. Robots often integrate rigid frameworks with soft materials that act as “skin”, typically embedded with sensing capabilities to replicate human sensory functions. These sensors enable robots to perform a wide range of tasks and navigate uncertainties by perceiving and responding to their environments. Especially in companion and healthcare robots, a soft surface is an important feature that enhances both safety and functionality [16]. However, the ability of soft robots to fully mimic these complex functions remains in the early stages of development [17].

Robotic skin sensors have emerged as a transformative technology in healthcare and rehabilitation, providing enhanced sensory feedback for prosthetic limbs, exoskeletons, companion robots, and healthcare robots [18]. By mimicking the sensory functions of human skin, robotic skins allow these devices to interact with their environment more naturally and effectively, improving operational efficiency and user experience [11]. Sensing plays an important role in the field of soft robotics, allowing robots to mimic human-like capabilities more effectively. Robots used in the healthcare field can integrate various sensors, such as tactile/pressure, temperature, and proximity sensors, as shown in Figure 1. These sensors enhance functionality and applicability in healthcare. Table 1 outlines the sensors used in the robots, along with their functions and associated challenges.

**Table 1 sensors-25-02895-t001:** The most common sensors used in robots.

Sensors	Functions	Challenges
Tactile and pressure sensor [19,20,21,22,23,24,25,26,27]	Touch/contact sensor—senses an object’s presence or absence; Force/pressure sensor—measures forces, including normal and shear forces	Cost; tactile sensor arrangement; wireless communication and crosstalk; software in real applications; modularization design; and transportability
Temperature/thermal sensor [28,29,30,31]	Magnitude and directions. Detects the surrounding temperature changes	Wireless communication and crosstalk; poor air permeability; incompatibility; instability; complex structural design; and costly manufacturing process
Proximity sensor [22,32,33,34,35]	Non-contact detection of objects/detecting nearby objects	Crosstalk; high cost; and large size

**Figure 1 sensors-25-02895-f001:**
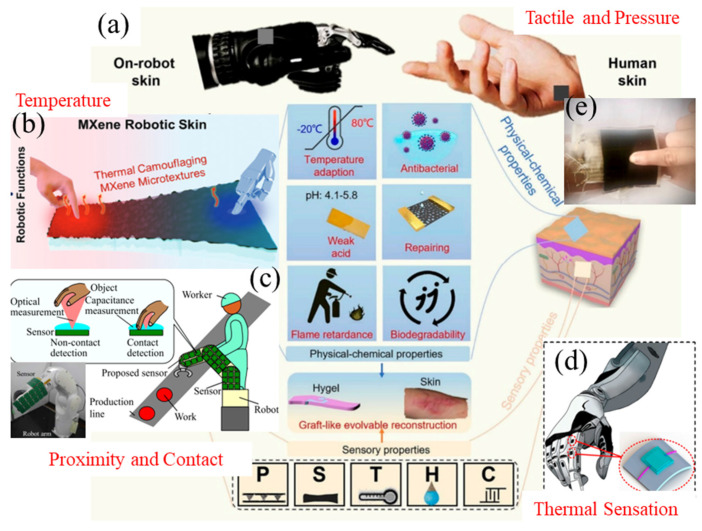
The most commonly used sensing technologies integrated into robots: (**a**) Electronic skin for robotics and wearables [36], (**b**) temperature sensor used for robot [37], (**c**) proximity and contact sensor for human cooperative robot [38], (**d**) thermal sensation [39], (**e**) pressure robotic sensor [27]. Reproduced with permission.

### 2.1. Tactile- and Pressure-Sensing Skins

Robotic skins capable of measuring pressure during physical contact can offer feedback to the robotic control system, improving the safety of interactions, especially when handling delicate objects, such as interacting with older adults. Tactile sensors are used in robotics and healthcare applications for tasks such as gripping, handling, object identification, autonomy, minimally invasive procedures, rehabilitation, etc. [40]. Therefore, pressure and tactile sensing play an important role in robotic and rehabilitation devices, particularly in prosthetics [41,42,43], where they are essential for enabling users to perceive and manipulate objects with greater precision. By detecting various forms of mechanical stimuli, such as pressure, texture, and vibration, these sensors mimic the sensory feedback of natural skin [44,45]. In the context of robotic skins, pressure and tactile sensing are even more sophisticated, engineered to detect a broader range of mechanical stimuli, including contact forces, shear forces, and vibrations [46]. These sensors enable robots to discern not only the texture, shape, and firmness of objects but also to respond to subtle changes in their environment, closely mimicking the nuanced tactile sensations experienced by humans. Examples of tactile sensors used in robotics are shown in Figure 2.

Tactile sensors are a fundamental component of robotic skins, providing the ability to detect and measure pressure, texture, and other mechanical stimuli [54]. These sensors are essential for a wide range of applications in healthcare and rehabilitation:Rehabilitation Devices: Tactile sensors allow devices such as robotic gloves or exoskeletons to monitor the amount of force a patient is applying during exercises [55,56]. This feedback is crucial for ensuring that the patient is performing movements correctly and safely, helping to guide therapy and track progress. For example, in hand therapy, tactile sensors can measure the pressure exerted by each finger, enabling precise adjustments to the therapy regimen based on real-time data [57];Companion Robots: Tactile sensors are key to making interactions with humans feel natural and comfortable. By detecting the pressure and texture of a human touch, these sensors enable the robot to respond appropriately, whether by offering a gentle hug, holding a hand, or providing a reassuring pat [58,59]. This ability to sense and react to touch enhances the robot’s role as a therapeutic tool, particularly in providing emotional support to individuals who may be lonely, anxious, or stressed;Healthcare Robots: Tactile sensors allow robots to interact delicately with patients and sensitive materials, such as handling medical instruments or assisting in patient transfers. The ability to detect the force being applied ensures that the robot can perform tasks safely without causing discomfort or injury to patients [60,61]. This is especially important in delicate procedures where precision is paramount.

The most commonly used technologies, as shown in Table 2, include capacitive, piezoresistive, and piezoelectric sensors. These sensors provide the vital feedback necessary for users to adjust their grip strength and fine-tune their movements, ultimately enhancing their ability to perform complex tasks confidently and accurately. This feedback significantly enhances users’ ability to perform complex tasks confidently and accurately, ultimately improving rehabilitation outcomes.

Table 2 provides an overview of three widely used robotic skin technologies: capacitive, piezoresistive, and piezoelectric sensors. The table details key performance metrics, including sensing mechanisms, measurement range, sensitivity, and response time, along with the advantages and disadvantages associated with each technology. Capacitive sensors generally offer high sensitivity and low power consumption. Piezoelectric sensors excel in dynamic pressure detection, providing fast response times; however, they require specific mechanical deformation for operation. Piezoresistive sensors are relatively simple and cost-effective, delivering good linearity, but their performance is often affected by temperature fluctuations.

By integrating these advanced sensing technologies, robotic skins can provide robots and prosthetic devices with the vital feedback needed to interact with the physical world in a more human-like manner and responsively. This integration enhances the functionality and adaptability of rehabilitation devices and significantly improves the user’s experience by providing a more natural and intuitive interface for controlling prosthetics and other assistive technologies. The development and refinement of pressure- and tactile-sensing skins continue to push the boundaries of what is possible in both robotic and rehabilitative applications, paving the way for more advanced, responsive, and human-centric devices that can better assist individuals in their daily lives.

### 2.2. Temperature-Sensing Skins

In healthcare, temperature-sensing skin enhances the functionality and realism of prosthetic limbs by enabling users to perceive warmth or cold, while companion robots simulate a comforting touch [76], thereby improving emotional and social support for patients [77]. Furthermore, in therapeutic devices, temperature-sensing skins can be used to monitor and regulate the temperature during treatments, ensuring that the patient receives the appropriate thermal stimuli. This is particularly relevant in heat or cold treatment therapy, where precise temperature control is essential for effectiveness and safety [78]. Incorporating thermistors and infrared sensors allows these skins to perceive temperature changes, adding a layer of sensory feedback that closely mimics the natural human experience [79]. For people using prosthetic limbs, the ability to sense warmth or cold can be crucial for tasks that involve temperature-sensitive materials or environments, such as cooking or handling heated objects.

The integration of temperature-sensing technologies into robotic and healthcare applications represents a significant advancement in creating more responsive and adaptive systems, as shown in Figure 3. By providing robots and prosthetic devices with the ability to detect and respond to thermal changes, these skins not only enhance the functionality and versatility of the devices but also contribute to a more intuitive and natural user experience [29]. As research and development in this area progress, temperature-sensing skins are anticipated to play an increasingly significant role across diverse domains, including industrial robotics and personalized healthcare.

Temperature sensors embedded within robotic skins enable the perception of thermal changes, adding another layer of sensory feedback that is particularly valuable for healthcare and companion robots.

Temperature-sensing skin enables robots to detect thermal properties in their environment. Li et al. [80] highlight that despite advancements, significant challenges still persist in adapting rigid thermal camouflaging technologies to the soft robots’ flexible and deformable bodies, particularly in healthcare applications, where thermal feedback can enhance the realism of prosthetic limbs or aid in therapeutic devices. These skins incorporate thermistors and infrared sensors to monitor temperature changes, allowing users to perceive warmth or cold. This can be crucial for tasks involving temperature-sensitive materials or environments where precise temperature control and monitoring are vital. The ability to sense temperature enables robots to handle tasks with greater precision and safety, ensuring they can adapt to varying thermal conditions in real time.

### 2.3. Proximity-Sensing Skins

Safety is imperative and must always be the top priority when robots interact with humans. Proximity and motion-sensing skins allow robots to detect objects at a distance without contact, enabling safer navigation and interaction in environments [81]. This capability is particularly crucial for robots operating near humans, where maintaining a safe and responsive interface is essential. Technologies used for proximity detection include optical sensors [82,83,84], infrared sensors [85,86], and capacitive sensors [87,88], which collectively enhance the robot’s awareness of its surroundings. Soft proximity sensing will be more effective because capacitive sensors offer touchless sensing capabilities, which are ideal for detecting the presence of humans or delicate objects. By integrating these technologies, robots gain enhanced spatial awareness, enabling them to operate safely and efficiently in environments shared with humans, such as healthcare facilities or homes [89]. This increased awareness not only prevents collisions but also allows robots to interact with their surroundings in a more adaptive and intuitive manner. For example, in healthcare settings [90,91], these sensors enable robots to navigate crowded spaces and avoid obstacles, ensuring safe and efficient interactions with patients and medical staff [92].

The importance of proximity and motion sensing in robotics cannot be overstated, especially in scenarios where robots must operate autonomously or semi-autonomously in environments shared with humans. Whether in a factory setting, healthcare facility, or home, robots equipped with these sensors can adjust their actions based on real-time data about their surroundings, ensuring that they can navigate complex spaces safely and efficiently [93].

By integrating multiple sensing technologies, proximity-sensing skins provide robots with a comprehensive understanding of their surroundings [94]. This enhanced spatial awareness allows robots to navigate through crowded or cluttered environments more safely and efficiently, avoiding obstacles and minimizing the risk of collisions. In environments where robots work alongside humans, such as in collaborative manufacturing, healthcare, or domestic settings, the ability to detect and respond to nearby people is critical for safety. Proximity sensors enable robots to slow down, stop, or alter their paths when a human approach reduces the risk of accidents and makes interactions more predictable and secure.

Robots operating in dynamic environments, such as warehouses, hospitals, or homes, must constantly adjust their movements in response to changing conditions. Proximity and motion-sensing skins allow robots to detect and react to moving objects or people in real time, thereby ensuring smooth and efficient navigation [88]. Proximity sensors also allow robots to interact intelligently with objects in their environment. For example, a robot equipped with these sensors can approach an object without making contact, assess its position and orientation, and decide how best to manipulate it [90]. This capability is essential for tasks that require precision, such as picking and placing items in manufacturing or handling fragile objects in healthcare settings.

Proximity and motion-sensing skins significantly enhance the capabilities of robots, making them more aware of their surroundings and better equipped to operate safely and effectively in environments shared with humans. As presented in Table 3, the incorporation of advanced sensing technologies, including capacitive, triboelectric, and magnetic sensors, equips robotic skins with enhanced capabilities for navigating complex environments, avoiding obstacles, and interacting with objects and humans in a more adaptive and intuitive manner. As sensor technologies continue to evolve, robots are expected to become more capable and versatile, enabling broader integration into various aspects of daily life.

## 3. Materials and Configurations in Healthcare and Rehabilitation Robotic Skins

In healthcare and rehabilitation applications, robotic skins need to be made from biocompatible materials that ensure safety, comfort, and durability [10,105,106]. The materials are very important for the effective functioning of assistive devices and companion robots due to their need to interact directly with the user’s skin and other surfaces without causing irritation or harm. The choice of materials is guided by several key considerations, including biocompatibility, flexibility, durability, and ease of maintenance, all of which are critical for ensuring the effectiveness and longevity of robotic skins in these sensitive environments.

Robotic skins are increasingly being made from stretchable and conformable materials. In rehabilitation, robotic skins need to comply with irregular surfaces, such as the human body or jointed exoskeletons, to function effectively. Stretchable and conformable skins, made from materials like elastomers [107,108,109] or hydrogels [110,111], provide the necessary flexibility to maintain close contact with these surfaces while delivering accurate sensory data. These materials are designed to accommodate the complex shapes and movements associated with the human body, maintaining a snug fit even as the underlying surface changes during motion. The elasticity and stretchability of these materials are important for ensuring that robotic skin can adapt to various forms and sizes, enabling it to function effectively across a wide range of rehabilitation devices. By enabling responsive interactions, these materials can significantly enhance the effectiveness of rehabilitation devices in supporting patient recovery.

Robots have irregular surfaces, and the stretchable and conformable skins can maintain close contact with irregular surfaces, which can provide the following advantages, as shown in Table 4:

The development of stretchable and conformable robotic skins has led to significant advancements in rehabilitation technology. By utilizing materials such as elastomers and hydrogels, these skins offer the flexibility and adaptability needed to interface effectively with the human body and other irregular surfaces, all while delivering accurate and continuous sensory feedback. This capability is critical for ensuring that rehabilitation devices can provide the necessary functionality and comfort, ultimately enhancing the user’s experience and improving therapeutic outcomes.

### Biocompatible and Flexible Materials and Sensor Configurations

Biocompatibility is one of the concerns when selecting materials for robotic skins for healthcare and rehabilitation. The materials used must not cause allergic reactions, irritation, or adverse effects when in contact with human skin [116,117]. This is particularly important for devices that are worn or used for extended periods, such as prosthetics, exoskeletons, and companion robots.

Silicone-based elastomers are one of the most commonly used materials in robotic skins due to their excellent biocompatibility, flexibility, and durability [118]. Elastomer materials are soft and comfortable against the skin, making them ideal for applications where the device must conform closely to the body, such as for prosthetic limbs or wearable rehabilitation devices [107,108]. Silicone elastomers are also resistant to extreme temperatures and environmental conditions, ensuring that they remain safe and effective over time.

Conductive polymers have also been used in robotic skins to provide both flexibility and the ability to conduct electrical signals, which is essential for sensory feedback [119,120,121]. These materials can be engineered to be biocompatible, allowing them to be safely used in close contact with skin. Conductive polymers are particularly valuable in applications where the robotic skin needs to integrate sensors that detect pressure, temperature, or other stimuli, enabling the device to interact intelligently with the user and their environment [122,123,124].

For companion robots, the materials used in robotic skins must be safe and enhance the user experience by being soft, pleasant to touch, and emotionally supportive. These robots are designed to provide comfort and companionship, and the tactile qualities of their skins play a significant role in achieving this goal.

Materials such as silicone-based elastomers and soft conductive polymers are selected for their ability to mimic the softness and flexibility of human skin. This makes the robot’s touch more natural and comforting, which is crucial in therapeutic settings where the robot is used to alleviate loneliness, reduce stress, and provide emotional support. The tactile experience is enhanced when the skin is soft to the touch, allowing the user to feel more easily and connect to the robot.

In addition to being soft, the materials used in companion robots’ skins are often chosen for their aesthetic qualities, such as color, texture, and warmth. These properties contribute to the overall sensory appeal of the robot, making it more approachable and engaging for users. For instance, a warm, smooth surface may be more inviting for a patient to touch, reinforcing the emotional connection between the robot and the user.

Current robots typically feature hard surfaces, raising challenges in integrating soft sensors of varying thicknesses. Understanding how these sensors affect the robot’s performance is crucial for seamless integration and enhanced functionality [125]. Zhu’s research [126] has investigated the relationship between substrate and sensor performance, employing both simulation and experimental analyses to elucidate the impact of varying structural configurations on sensor sensitivity and touch force response. Traditional robotic systems, typically built with rigid frames and integrated sensors, are increasingly being replaced by soft robotic platforms enabled by the use of biocompatible and flexible materials. This transition is generally achieved through three different structural configurations, each designed to meet specific functional objectives while introducing certain limitations. In many applications, the direct integration of soft sensors onto rigid robotic configuration structures enables high-resolution, real-time interaction with the environment and is commonly employed in systems such as robotic hands and exoskeletons. However, the rigidity of the underlying frame may limit overall flexibility and reduce user comfort. Traditional robotic systems with rigid frames, when combined with compliant substrates and soft sensors mounted on top, enhance surface conformity, mechanical flexibility, and sensitivity. This configuration is particularly advantageous for applications such as tactile sensing skins, stretchable electronics, and bioinspired robotics, where improved surface sensing, flexibility, and responsiveness are essential for optimal performance. Furthermore, rigid sensors embedded beneath soft substrates provide increased mechanical protection and adaptability at the interface, making them especially suitable for prosthetic devices and biologically inspired robotic systems. However, the inherent rigidity of the embedded sensors may limit the system’s ability to conform to irregular anatomical surfaces, potentially affecting signal fidelity [127,128,129,130,131]. Hence, it is crucial to integrate biocompatible and flexible materials more effectively into these structural configurations to improve the safety, responsiveness, and functional performance of next-generation soft robotic systems, particularly in the fields of healthcare and assistive technologies.

The integration of sensors into a material is a complex process that requires the skin to remain flexible and responsive while housing delicate electronic components. The materials must support the consistent transmission of electrical signals from the sensors to the processing units, ensuring accurate and reliable feedback. Conductive polymers and other advanced materials are used to achieve this integration without compromising the flexibility or durability of the skin.

Over time, repeated use, cleaning, and exposure to environmental factors can affect the sensory capabilities of robotic skin. The chosen materials must be able to protect the embedded sensors from damage, ensuring that the skin continues to provide accurate feedback. This longevity is essential for the device’s continued effectiveness in therapeutic and assistive roles. The development of robotic skins for healthcare and rehabilitation necessitates careful consideration of material selection and structural configuration to ensure biocompatibility, comfort, durability, and effective sensory performance.

## 4. Applications in Healthcare and Rehabilitation

Robotic skin plays a crucial role in enabling more natural and responsive interaction between machines and humans, with wide-ranging applications in prosthetics, rehabilitation, and healthcare. As discussed in previous sections, materials such as silicone elastomers and hydrogels, combined with sensor technologies like capacitive, piezoresistive, and piezoelectric sensors, form the foundation of these systems. This section provides an overview of how these technologies are being utilized in healthcare and rehabilitation contexts to improve performance and user outcomes.

Robotic skin technologies offer a wide array of applications across healthcare and rehabilitation fields. Due to pressing challenges such as aging populations, healthcare workforce shortages, and persistent gaps in accessibility, there is a growing need for innovative solutions to support and transform healthcare delivery. As illustrated in Figure 4, robotics presents promising opportunities across several critical domains that warrant focused development. These include robot-assisted living for the elderly and disabled, interventional rehabilitation to enhance patient recovery, and robotic telemedicine to bridge geographical barriers to healthcare access. Additionally, robotics can play a crucial role in disease prevention, personalized health management, and automated first aid responses. By targeting these areas, the integration of robotics into healthcare systems can help address current limitations while promoting more efficient, equitable, and responsive care models. Nevertheless, current robotic systems predominantly feature rigid surface architectures, which may constrain their adaptability and suitability in dynamic or patient-sensitive environments. Therefore, the development and adoption of soft robotic skin hold significant potential for future advancements, offering a safer and more adaptable solution for assisting individuals in healthcare settings.

### 4.1. Prosthetics

Robotic skins are used in prosthetics to provide sensory feedback to users, improving their ability to interact with the environment [133]. By integrating advanced sensing technologies into a flexible, skin-like material, these robotic skins replicate some of the sensory functions of natural skin, allowing prosthetic limbs to convey critical information about touch, pressure, and texture to the user [134]. This sensory feedback is crucial for enabling more precise control of prosthetic devices, leading to more natural and effective movements [135]. These skins can also be applied to robotic exoskeletons, where they enhance the interaction between the robot and the human body [136].

In prosthetics, robotic skins allow users to regain a sense of touch that is otherwise lost with traditional prosthetic limbs [137]. These skins are embedded with various sensors that detect mechanical stimuli, such as pressure, shear forces, and vibrations, and convert them into electrical signals that are relayed to the user’s nervous system directly or through other sensory modalities [138]. This feedback enables users to adjust their grip strength, discern the texture of objects, and perform delicate tasks with greater confidence and accuracy.

Robotic skins offer several advantages in prosthetics, improving the overall user experience. First, they enhance tactile sensation, enabling users to perceive touch and engage in delicate tasks, such as handling small or fragile objects. Real-time feedback from these skins helps users make more informed decisions during interactions. Additionally, robotic skins improve motor control by offering sensory feedback, allowing users to fine-tune their grip strength and movements to achieve smoother and more natural actions. Furthermore, the enhanced sensory experience increases user comfort and confidence, reducing the mental effort required to use the prosthetic limb and making it feel more like a natural body extension.

Integrating robotic skins into exoskeletons enhances the interaction between the device and the human body. These skins provide sensory feedback that informs both the user and the exoskeleton’s control system about applied forces and movements, enabling the device to adapt more precisely to the user’s motions for better, more comfortable assistance. Their ability to detect pressure and shear forces prevents excessive force from being applied, reducing the risk of discomfort or injury, especially during long-term or intensive use [127]. This adaptability ensures that the exoskeleton can provide the right amount of assistance based on the user’s needs at any given moment, whether walking, lifting, or performing rehabilitation exercises.

In conclusion, the integration of robotic skin technologies into rehabilitation and assistive devices such as prosthetics and exoskeletons offers substantial benefits. These systems enable more accurate monitoring of patient progress, allowing for adaptive and personalized therapeutic strategies that may enhance recovery outcomes. In prosthetic applications, robotic skins contribute to restoring tactile feedback and improving motor control, thereby increasing both functionality and user satisfaction. Furthermore, the enhanced human–machine interface provided by these technologies improves the overall safety, comfort, and effectiveness of assistive devices. As research and development in robotic skins continue to evolve, their role in improving the quality of life for individuals requiring rehabilitation and assistive technologies is expected to expand significantly.

### 4.2. Companion Robots

Companion robots equipped with advanced robotic skins play an increasingly vital role in therapeutic settings by providing emotional and social support to patients across various conditions. These robots are specifically designed to interact with humans in a natural and comforting way, using their sensory capabilities to simulate warmth, respond to touch, and even mimic human-like gestures [139,140,141]. This ability to engage with patients more intimately and personally makes them highly effective tools in alleviating loneliness, reducing stress, and improving overall mental well-being [142,143].

The integration of robotic skins into companion robots is crucial to their ability to perform their therapeutic functions effectively [143]. These skins are embedded with sensors that allow the robot to detect and respond to various forms of physical contact, such as gentle touches or hugs [142]. This sensory feedback is essential for creating authentic and emotionally resonant interactions, which are key to the success of these robots in therapeutic contexts [139].

Robotic skins enable companion robots to detect touch and respond in ways that simulate empathetic human interactions. For example, a gentle pat might prompt the robot to move closer or make a comforting gesture, helping patients feel understood and cared for [144]. This responsiveness fosters a meaningful connection, which is crucial in therapeutic settings. Additionally, robotic skins can simulate warmth, providing a soothing, reassuring sensation that helps reduce loneliness and promote security, especially for isolated patients. Beyond touch, these robots can mimic human-like gestures, such as holding hands or offering hugs, enhancing non-verbal communication [145]. This is especially beneficial for patients with dementia, autism, or other conditions where such interactions are vital for therapy [59].

Companion robots with robotic skins offer therapeutic benefits by addressing the emotional needs of patients, particularly those in elderly care or those with chronic illnesses [146,147]. First, they help alleviate loneliness by providing consistent empathetic interactions that simulate genuine companionship and reduce the emotional toll of isolation. The tactile feedback from robotic skins enhances this sense of connection. Second, these robots’ physical touch can lower stress and anxiety, creating a calming effect, which is especially valuable in hospital or nursing home environments. Additionally, their ability to engage patients in meaningful interactions improves their mental well-being by boosting mood and encouraging positive emotions. Finally, by offering emotional support and comfort, companion robots can enhance therapy engagement, leading to better therapeutic outcomes [148,149].

Companion robots equipped with robotic skins are making significant strides in therapeutic settings by offering patients the emotional and social support they need. The sensory capabilities of these skins are central to creating interactions that feel natural, warm, and human-like, thereby enhancing the therapeutic impact of robots. As these technologies continue to evolve, we expect companion robots to play an even more prominent role in supporting mental and emotional well-being across various therapeutic contexts.

### 4.3. Therapeutic Devices

Robotic skins are increasingly being integrated into therapeutic devices designed for rehabilitation, where they play a crucial role in providing real-time feedback that enhances the effectiveness of therapy [150,151]. These advanced skins, embedded with various sensors, offer valuable data that helps patients and therapists monitor progress, adjust treatments, and achieve better outcomes [142,152]. There are many applications of robotic skins in devices, such as the following:Pressure Sensors in Robotic Gloves [153,154,155]: One of the most prominent applications of robotic skins in rehabilitation is in robotic gloves used for hand therapy exercises. These gloves are equipped with pressure sensors that can measure the force applied by the patient during exercise and provide detailed feedback on their performance. For instance, during a grip-strengthening exercise, the sensors can detect the amount of force exerted by each finger, allowing therapists to assess the patient’s strength, coordination, and progress over time. These data are invaluable for tailoring therapy to the patient’s specific needs, identifying areas of improvement, and making necessary adjustments to the treatment plan;Monitoring and Adjusting Therapy [138,156]: The feedback provided by robotic skins in therapeutic devices allows for continuous monitoring of patient progress. Therapists can use this information to adjust the intensity, frequency, and type of exercises prescribed, ensuring that the therapy remains aligned with the patient’s capabilities and goals. For example, if a patient improves in hand strength, the therapist might increase the exercise’s resistance level to challenge the patient further. Conversely, if the sensors indicate that the patient is struggling, the therapist can modify the exercises to prevent overexertion and reduce the risk of injury;Interactive Rehabilitation with Companion Robots [157,158]: Robotic skins are also integrated into companion robots that offer interactive exercises designed to physically and emotionally engage patients. These robots can guide patients through various therapeutic activities, such as stretching, lifting, or balancing exercises, while providing real-time feedback through their sensory capabilities. These robots can create a more immersive and motivating therapy experience by responding to the patient’s movements and touch. Moreover, the interactive nature of these exercises helps keep patients engaged, which is particularly important in long-term rehabilitation, where maintaining motivation can be challenging.

In rehabilitation, robotic skins offer numerous benefits, including the following:Personalized Therapy [159]: The data collected by sensors in robotic skins allow for highly personalized therapy, where exercises and treatment plans are tailored to teach the patient’s individual needs and progress. This level of customization is critical for achieving optimal rehabilitation outcomes as it ensures that the therapy is neither too easy nor too difficult for the patient;Objective Progress Tracking [160,161]: Robotic skins provide objective, quantifiable data on patient performance, which is essential for tracking progress over time. These data can be used to create detailed progress reports, helping patients and therapists see the improvements made and identify areas needing further attention. This objective tracking also supports evidence-based adjustments to therapy, thereby making the rehabilitation process more efficient and effective;Enhanced Patient Engagement [160,162]: The interactive capabilities of robotic skins, particularly when integrated into companion robots, help to enhance patient engagement during rehabilitation exercises. Making therapy more interactive and responsive makes patients more likely to stay motivated and committed to their rehabilitation program. The emotional connection fostered by companion robots can also reduce feelings of frustration or isolation, which sometimes accompany long-term rehabilitation;Improved Therapy Outcomes [163]: Ultimately, the integration of robotic skins into therapeutic devices contributes to improved therapy outcomes. The ability to provide real-time feedback, adjust treatments based on objective data, and engage patients more effectively all combine to create a more effective rehabilitation process. Patients can achieve better results in a shorter amount of time, which is particularly beneficial for those recovering from injuries or surgeries where time-sensitive recovery is crucial.

The integration of robotic skins into therapeutic devices for rehabilitation purposes represents a significant advancement in the field of physical therapy. By providing detailed feedback and enabling personalized interactive therapy, these technologies enhance the effectiveness of rehabilitation, helping patients recover more efficiently and successfully. As advancements in this field continue, increasingly sophisticated applications of robotic skins in rehabilitation are anticipated, with the potential to further enhance patient care and therapeutic outcomes.

## 5. Challenges in the Development of Practical Applications

One of the challenges in developing robotic skins for healthcare, particularly in companion robots, is ensuring long-term durability [164]. These robotic skins are designed to interact closely with humans, often on a daily basis, which subjects them to a wide range of physical and environmental stresses [105,165]. As a result, the durability of these skins is paramount to their effectiveness and longevity in therapeutic and assistive roles.

The development of durable robotic skins for healthcare applications presents a complex set of challenges, particularly in ensuring that these skins can withstand the rigors of daily use, environmental exposure, and mechanical stresses [166]. Addressing these challenges requires a multidisciplinary approach that combines advanced materials science, sensor engineering, and rigorous testing. By overcoming these durability issues, developers can create more reliable and long-lasting robotic skins that enhance the effectiveness and safety of companion robots, prosthetics, and other healthcare devices, ultimately improving patient outcomes and quality of life.

The integration of sensors into robotic skins is critical for enabling robots to interact intelligently and sensitively with their environments. However, these sensors, along with the processing units that interpret their data, require a continuous supply of power to function effectively [167]. Managing the energy consumption in these systems without compromising their performance is one of the most significant challenges in the development of robotic skins, particularly for portable devices such as prosthetics, exoskeletons, and companion robots.

The challenge of managing the energy consumption in robotic skins is a multifaceted problem that requires innovative solutions in sensor design, power management, and energy harvesting. By developing low-power, energy-efficient sensors and optimizing power management systems within these devices, it is possible to create robotic skins that deliver high performance while minimizing the need for frequent recharging. These advancements are crucial for ensuring that they remain reliable, responsive, and user-friendly over extended periods of use. Table 5 summarizes the key challenges in robotic skin development and provides some strategies for addressing them and their potential implications for scalability, cost, and performance.

In healthcare applications, comfort and biocompatibility are paramount considerations for the development of robotic skins [173]. These skins are often used in devices that come into direct contact with the human body, which means they must be safe, comfortable, and suitable for extended use. Achieving the balance between comfort, flexibility, and functionality is a key challenge in the material selection and design of robotic skins, particularly for companion robots intended to provide emotional support.

Companion robots are often used in close, prolonged contact with humans, where tactile experience plays a significant role in the effectiveness of the interaction [5]. For a companion robot designed to offer hugs or hold a patient’s hand, it must have a skin that feels soft, warm, and non-irritating to provide genuine comfort and emotional reassurance. Advancements in comfort and biocompatibility will particularly benefit companion robots designed to provide emotional and social support.

Ensuring comfort and biocompatibility in robotic skins is a complex challenge that requires careful consideration of material properties, design, and user interaction. By focusing on advanced material selection, ergonomic design, and personalized solutions, developers can create robotic skins that provide the necessary functionality and offer a comfortable, safe, and enjoyable user experience. This is particularly important in healthcare applications, where the well-being and comfort of the user are paramount, and companion robots are expected to provide both physical and emotional support through their interactions.

Robotic skins, integral to healthcare and companion robots, must operate in complex and dynamic environments. These environments often generate noisy data, which can interfere with the accuracy of sensing and feedback, ultimately affecting the robot’s ability to interact meaningfully with its surroundings [94]. In companion robots, where touch sensitivity is crucial for providing emotional and therapeutic support, inaccurate or delayed responses due to noisy data can undermine the robot’s effectiveness. Therefore, employing effective signal processing techniques is essential to filter out irrelevant information and ensure that robotic skins deliver reliable and meaningful sensory data. Effective signal processing is crucial for managing noisy data and ensuring that the robotic skins provide accurate and meaningful sensory feedback. Signal processing techniques help filter out irrelevant data, reduce noise, and enhance the clarity of sensory inputs essential for the robot’s functionality.

Accurate sensing and response to touch are critical for the therapeutic role of companion robots [1]. These robots often interact closely with patients and offer comfort and emotional support through physical contact. Inaccurate responses due to noisy data can disrupt these interactions, potentially causing frustration or discomfort for the user. By employing advanced signal processing techniques, companion robots can ensure that their responses to touch are precise and appropriate, thereby enhancing their effectiveness as therapeutic tools.

In rehabilitation settings, robotic skins equipped with signal processing capabilities can provide more accurate feedback during exercises, helping therapists monitor patient progress and adjust treatments accordingly. Reliable sensory feedback is essential for prosthetics and exoskeletons to enable users to control their movements effectively [119]. Signal processing ensures that the data from the sensors embedded in the robotic skin are clear and actionable, allowing users to perform tasks with greater precision and confidence. This is particularly important in dynamic environments where the risk of interference and noise is high.

The complex environments in which healthcare and companion robots operate often generate noisy data that can interfere with accurate sensing and feedback. Effective signal processing is essential for filtering out irrelevant information, ensuring that robotic skins provide reliable and meaningful sensory data. By employing the techniques outlined in this review, including noise filtering, sensor fusion, adaptive filtering, machine learning algorithms, and redundancy, developers can enhance the performance of robotic skins, ensuring they satisfy the requirements of healthcare applications. This is particularly important for companion robots, where accurate and responsive touch interactions are vital for fulfilling their therapeutic role and providing emotional support to users.

Another challenge is the production of affordable and scalable robotic skins for mass manufacturing. Current manufacturing processes are often costly and lack scalability, limiting their widespread adoption, which is an aspect frequently overlooked in many research studies and reviews. While robotic skins have significant potential in healthcare, rehabilitation, and companion robots, the existing techniques are too complex and expensive to meet the affordability and scalability required for mass production.

Addressing these challenges is essential to bring robotic skin technologies to the market, enabling broader application. Innovative solutions in materials science, manufacturing techniques, and design are necessary to make robotic skins more affordable and scalable, unlocking their potential for widespread use. As efforts to reduce costs and improve scalability continue, robotic skins are poised to become a key component of next-generation healthcare devices, offering new opportunities to enhance quality of life and personalized care.

## 6. Future Directions in Healthcare and Rehabilitation

As robotic skin technology continues to evolve, its potential applications in healthcare and rehabilitation are expanding, with promising transformative improvements in patient care, therapy outcomes, and overall quality of life. The following sections explore the key areas where advancements in robotic skin technology are expected to drive future developments.

### 6.1. Advanced Material Development

The future of robotic skin development in healthcare and rehabilitation hinges on the advancement of materials that are durable, flexible, biocompatible, and capable of adapting to various conditions. Researchers are actively exploring the potential of self-healing materials, such as polymers [123], that can repair themselves after damage, extending the lifespan of robotic skins and reducing maintenance costs. Such materials are particularly valuable in companion robots that frequently engage in tactile interactions where wear and tear are inevitable.

Additionally, polymers [124] can change their properties in response to external stimuli such as temperature. These materials allow robotic skins to adjust their stiffness, texture, or conductivity based on the specific needs of the user or environment, thereby enhancing both comfort and functionality. The polymer skin could become more rigid to provide support during physical therapy exercises or soften to increase comfort during rest periods.

Therefore, the development of biocompatible and environmentally friendly materials is also crucial. As robotic skins are used in devices that come in direct contact with human skin, materials must be safe, non-toxic, and capable of long-term use without causing irritation or allergic reactions. Innovations in bio-based materials and sustainable manufacturing processes can lead to robotic skins that are not only effective but also environmentally responsible.

### 6.2. Integration with Artificial Intelligence

Artificial intelligence (AI) is poised to play a pivotal role in the development of next-generation robotic skins, particularly in enhancing data processing and adaptive learning capabilities. Robotic skin systems generate extensive, multimodal sensory data, such as tactile input, pressure, and temperature, that require intelligent analysis for meaningful interpretation. AI-driven systems enable robust handling of these data, thereby supporting more responsive, personalized, and context-aware interactions in healthcare environments. Among various AI techniques, machine learning approaches, particularly Convolutional Neural Networks (CNNs), have demonstrated considerable promise due to their ability to extract spatial features from complex, high-dimensional sensor data. CNNs facilitate the real-time recognition of tactile patterns, pressure distributions, and motion cues, all of which are critical for applications such as prosthetic control, rehabilitation monitoring, and interactive assistive robotics [181]. The integration of CNN-based models within robotic skin systems significantly enhances the functionality of assistive technologies, offering improved adaptability, increased precision, and user-specific feedback.

For rehabilitation devices [182], AI can be used to monitor patient progress in real time, adjusting therapy protocols based on the patient’s performance and feedback. Machine learning algorithms can identify patterns in data that indicate a patient’s readiness to advance to more challenging exercises or the need for a modified approach. This level of personalization could significantly improve rehabilitation outcomes, making the therapy more effective and responsive to individual needs. For example, Gao’s study on CNN models for recognizing body motion intent demonstrates their potential to enhance rehabilitation effectiveness through adaptive control and feedback [183].

In companion robots [184], AI integration allows for adaptive learning, where the robot can evolve its behavior based on the user’s emotional and physical states. By continuously learning from interactions, the robot can tailor its responses to provide more meaningful and supportive companionship. For instance, by employing CNN methods, a companion robot can detect signs of stress or anxiety through changes in touch patterns and adjust its behavior to offer calming interactions or encourage relaxation exercises [185,186,187].

With advances in robotic skin technology, the potential for creating personalized rehabilitation solutions and companion robots has grown. The ability to customize robotic skins to meet the specific needs and preferences of individual patients could revolutionize rehabilitation programs.

For rehabilitation, customizable skins can be designed to provide targeted therapy, such as pressure-sensitive areas that focus on strengthening specific muscle groups or skins that adjust their resistance based on the patient’s progress. This level of personalization can enhance the effectiveness of therapy by ensuring that each exercise is optimally tailored to the patient’s current capabilities.

In companion robots, personalization can extend beyond physical customization to include behavioral and emotional adaptations. By integrating AI with robotic skins, companion robots can learn and adapt to user preferences, moods, and emotional states, providing genuine personal and empathetic support. For example, a companion robot can adjust its touch, voice tone, or interaction style based on the user’s current mood, offering comfort, encouragement, or companionship.

### 6.3. Expanded Applications

The future of robotic skins is not limited to their current applications. As technology advances, these skins may find new uses in remote healthcare monitoring, wearable health devices, and chronic condition management.

Robotic skin can be integrated into remote monitoring systems that track vital signs and other health metrics, providing continuous, real-time data to healthcare providers. This could be especially beneficial for patients with chronic conditions who require ongoing monitoring but prefer to remain in their homes. The ability to detect subtle changes in a patient’s condition early could lead to more timely interventions, potentially preventing complications and improving outcomes.

In wearable devices, robotic skins can provide a more natural and comfortable interface for monitoring health metrics such as heart rate, temperature, and hydration levels. These skins could also offer haptic feedback, alerting the user to abnormal readings or prompting them to take specific actions, such as taking medication or performing exercises.

As the global population ages, the demand for technologies that support independent living and enhance the quality of life of older adults is increasing. Companion robots with advanced robotic skins can play a critical role in providing continuous care and support to aging populations. These robots can assist with daily activities, monitor health conditions, and provide emotional support while adapting to the users’ changing needs over time.

## 7. Conclusions

Robots are set to revolutionize healthcare and rehabilitation by offering innovative solutions, such as robots with skins, that can significantly enhance the functionality and user experience of assistive devices and companion robots. These advanced materials and systems have the potential to bring new levels of sensitivity, adaptability, and personalization to devices that support patients in their daily lives and therapeutic routines. Future research and innovations in robotic skin development will largely focus on advancements in advanced and sustainable materials combined with the integration of artificial intelligence and machine learning to enhance data processing and adaptive learning capabilities. These advancements will enable robotic skins to interpret sensory inputs with greater accuracy, thereby facilitating more refined and personalized interactions. Beyond the current applications, future developments are anticipated to extend the use of robotic skins to areas such as remote healthcare monitoring, wearable health devices, and chronic disease management. These emerging directions highlight the ongoing evolution of robotic skin technology, particularly in the domains of medical and assistive care.

Although challenges persist in areas such as durability, power management, and comfort, ongoing research is actively addressing these issues, paving the way for more advanced and effective robotic skins. As these challenges are overcome, this technology holds great promise for dramatically enhancing patient care and well-being. By offering more personalized and responsive solutions, robotic skins have the potential to transform healthcare and rehabilitation, leading to improved outcomes and quality of life for countless individuals.

In prosthetics, robotic skins equipped with pressure and tactile sensors can provide users with a sense of touch, allowing them to control their artificial limbs better and interact more confidently with their environment. In companion robots, integrating robotic skins can make interactions feel more genuine and supportive, thereby improving the emotional connection between the robot and the user. These advancements not only improve the practical utility of assistive devices but also enhance the overall user experience, leading to greater satisfaction and better outcomes in rehabilitation and daily living.

This paper presents a narrative review of current demands and advancements in robotic skin technologies within healthcare applications, with particular emphasis on sensing technologies, materials, and the critical challenges that must be addressed to support future development. The field remains dynamic and highly promising, driven by innovations in materials, such as soft elastomers and hydrogels, that closely mimic the tactile sensitivity and mechanical flexibility of human skin. These advancements have shown considerable potential in improving prosthetic devices, exoskeletons, and companion robots by enabling the perception of pressure, temperature, and motion, thereby facilitating more natural and intuitive interactions with the environment.

Despite these encouraging developments, several key challenges persist. Notably, there is a pressing need to enhance material durability, enable real-time processing of high-dimensional sensory data, and achieve seamless integration with complementary technologies such as artificial intelligence. The selection of materials must carefully balance flexibility, mechanical strength, and biocompatibility, particularly for long-term deployment in medical settings. Concurrently, AI techniques, especially those employing deep learning architectures such as Convolutional Neural Networks, are being refined to interpret complex tactile data, which is essential for robust and reliable performance in healthcare and rehabilitation contexts.

Looking forward, the convergence of AI, advanced sensing mechanisms, and material science holds substantial promise. As these technologies continue to mature, robotic skin systems are anticipated to become increasingly adaptive and responsive, contributing to enhanced prosthetic control and personalized therapeutic interventions. The long-term goal is to develop intelligent robotic skin that not only replicates the sensory and mechanical characteristics of human skin but also dynamically adapts to individual user needs and environmental changes.

Future developments will likely be driven by the convergence of advanced materials, intelligent sensing, and AI. As these fields continue to integrate, robotic skin systems are expected to become more responsive, adaptive, and capable of supporting advanced applications such as personalized prosthetic control and real-time therapeutic feedback. The development of intelligent robotic skin that mimics the functional characteristics of human skin is a key objective, with the potential to significantly advance healthcare technologies and improve human–robot interaction.

## Figures and Tables

**Figure 2 sensors-25-02895-f002:**
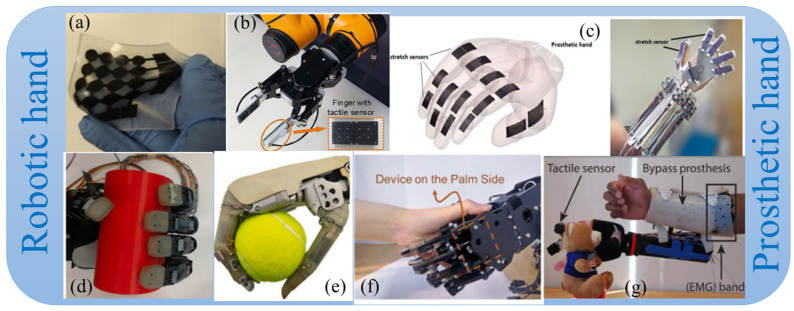
(**a**) Soft robotic sensor deforms under tensile forces [47]; (**b**) robot hand with tactile sensors. [48]; (**c**) stretch sensors for prosthetic hand [49]; (**d**) robotic hand with piezoresistive sensors [50]; (**e**) force-sensing grasp of a soft tennis ball [51]; (**f**) the contact force extremums in a hand-shaking operation [52]; (**g**) robotic hand controlled by EMG on a bypass prosthesis [53]. Reproduced with permission.

**Figure 3 sensors-25-02895-f003:**
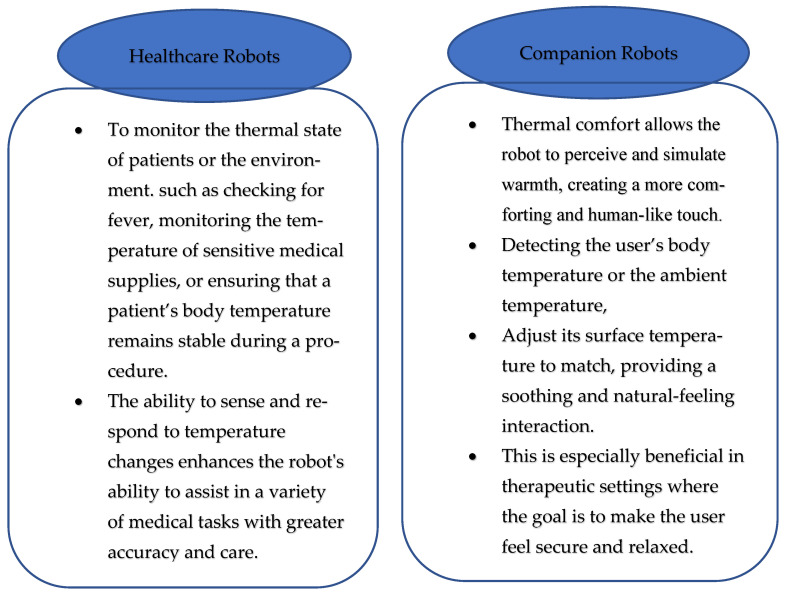
Temperature sensors used in healthcare and companion robots and their functionality.

**Figure 4 sensors-25-02895-f004:**
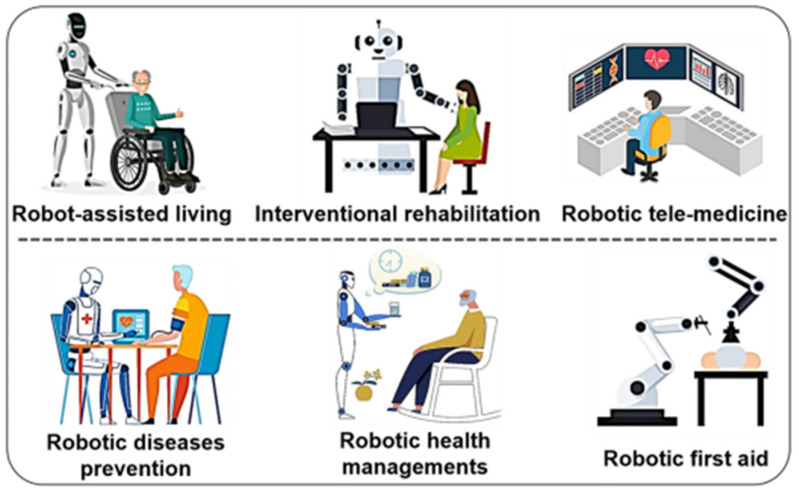
New technologies within healthcare robotics [132]. Reproduced with permission.

**Table 2 sensors-25-02895-t002:** Most commonly used tactile and pressure robotic skin sensing technologies.

Technologies	Capacitive [62,63,64,65,66]	Piezoresistive [67,68,69]	Piezoelectric [70,71,72,73,74,75]
Sensing Method	Change in capacitance Silicon-based	Change in resistance	Strain (stress) polarization
Measurement Ranges and Sensitivity	61.77%/N Pressure range 2–45 kPa S = 0.63 f F/kPa	From 0 to 800 kPa From 0.8 to 2 MPa	Linear force detection range of 1–11 N, 35.6 mV/N Detection range of 50 kPa, 0.5 V/N
Response Time	5 s	Less than 60 ms; 2.5 mm/s	119 ms
Advantages	High spatial resolution Good frequency response Long-term drift stability High sensitivity Low-temperature sensitivity Low power consumption	High spatial resolution Good sensitivity Low noise Low cost	Excellent resistance High mechanical strength Good plasticity Relatively high accuracy High power density High bandwidth High efficiency High-frequency response
Disadvantages	Stray capacitance Noise susceptible	Rigid and Fragile Severe hysteresis Higher power consumption Large hysteresis Low repeatability	Poor spatial resolution Dynamic sensing only

**Table 3 sensors-25-02895-t003:** Most commonly used proximity robotic skin sensing technologies.

Technologies	Sensing Principle	Sensitivity	Hysteresis	Advantages	Disadvantages
Capacitive [88,94,95,96,97]	Detect proximity through changes in capacitance	High	Low	Low cost, fast dynamic response, excellent sensitivity, good spatial resolution, large dynamic range	Interference from the surrounding environment (noise susceptible)
Triboelectric [98,99,100,101]	Generate electrical signals from mechanical contact	High	Moderate	Self-powered	Charge attenuation, susceptible to environmental interference
Magnetic sensor [102,103,104]	Detect proximity of metallic objects via changes in magnetic fields	High for detecting magnetic fields	Low	Not affected by non-magnetic objects	Restricted object detection

**Table 4 sensors-25-02895-t004:** The advantages of using stretchable and conformable skins.

Material Type	Elastomers	Hydrogels	
	Stretchable polymers (e.g., PDMS, and Ecoflex)	Soft polymer (e.g., polyacrylamide-based)	Advantages of Implementation
Enhanced Sensory Feedback [10,56,112]	Commonly used for embedding capacitive, piezoresistive, or triboelectric sensors	Natural ionic sensing; high sensitivity	Maintains contact between the robotic skin and the body or exoskeleton surface for accurate detection of physical stimuli such as pressure, shear, and temperature.
Improved Comfort and Usability [113,114]	Durable, easy to handle, easy to manufacture, mold, and attach to robotic frameworks	Skin-like softness; hydration dependent	Stretchable and conformable skin moves with the user’s body, reducing discomfort and irritation that could arise from rigid or poorly fitting materials.
Maintained Functionality During Movement [115]	High elasticity, reliable under stress, environmental resilience	Requires mechanical reinforcement, environmental sensitivity	The flexibility of these materials allows the robotic skin to maintain its functionality even as the user moves.

**Table 5 sensors-25-02895-t005:** The key considerations in robotic skin development.

	Key Challenges	Strategies	Implication
Durability and Longevity [168,169]	Material Fatigue: Mechanical stress causes cracks or degradation; Sensor Wear: Leads to data inaccuracy or failure; Environmental Damage: Heat, moisture, and sweat cause corrosion or swelling; Impact Effects: Surface wear reduces sensor performance.	Use durable materials to resist wear; Create long-lasting, accurate sensors; Make robotic skins environment-resistant and body-fluid-proof; Design skins to withstand abrasion and impact while staying flexible and functional.	
Power Consumption [170,171]	Power Demands of Sensors and Processing Units: Efficient power management is essential for the continuous operation of lightweight, portable devices and their processing units; Portability and Battery Life: Battery life is critical for portable devices (e.g., prosthetics and exoskeletons) and companion robots, which require constant power for sensors and actuators; Thermal Management: Managing heat from sensors and processing units is vital to prevent performance degradation, discomfort, and safety risks.	Robotic skin sensors (e.g., pressure, temperature, and proximity) must continuously collect and transmit data; Energy-efficient designs should minimize heat output while ensuring sensor accuracy and responsiveness.	Extended Operating Time: Effective energy management allows prosthetics, exoskeletons, and companion robots to operate longer without interruption, which is crucial for daily mobility, interaction, or therapy; Reliability and Responsiveness: Consistent power ensures robotic skins maintain sensor accuracy and responsiveness, meeting user expectations for reliable performance; User Comfort and Safety: Efficient power management reduces heat, enhancing comfort and safety by keeping device temperatures within safe limits.
Comfort and Biocompatibility [172,173,174]	Prolonged Contact: Materials must be soft, flexible, and conform to the body to avoid irritation, chafing, or pressure sores; Biocompatibility: Materials should be non-toxic and hypoallergenic to prevent adverse reactions with human tissue; Thermal Comfort: Effective heat management is essential to avoid overheating or excessive cold, ensuring user comfort; Moisture Management: Materials must manage sweat and humidity to prevent discomfort and skin irritation; Flexibility and Durability: A balance is needed, and materials should be flexible for comfort yet durable for support and protection	Materials must retain softness and flexibility despite repeated use and environmental exposure; It is critical for skin-worn devices that the materials must remain safe and non-degradable, even when exposed to sweat or body fluids; Materials should maintain a comfortable temperature during use; Breathable or moisture-wicking materials enhance long-term comfort; Materials must balance flexibility for natural movement with durability to withstand repeated use without degradation.	
Signal Processing and Noise [175,176,177,178]	Complex Environmental Interference: Healthcare and companion robots face unpredictable environments with electromagnetic signals, lighting changes, temperature shifts, and physical obstructions, which can cause noise in sensory data; Multiple Sensor Inputs: Robotic skins use various sensors (e.g., pressure and temperature) that can interfere with each other, leading to noisy or corrupted data; Variability in Human Interaction: Human touch varies in force, duration, and contact area, creating inconsistent data that are hard to interpret accurately.		
Affordability and Scalability [179,180]	Expensive Material: The costs of materials, like advanced polymers, elastomers, and hydrogels, are chosen for flexibility, durability, and biocompatibility; Complex Production: Production involves intricate processes requiring specialized equipment, such as multi-layer integration and sensor embedding; Customization Costs: Custom designs for specific applications add complexity and limit scalability; Sensor Integration and Testing: Precise sensor integration and rigorous testing increase production time and costs.	Producing robotic skins at a low cost for large-scale manufacturing is challenging; Ensuring uniform quality across large production runs is challenging, hindering scalability. Balancing personalized designs with mass production complicates accessibility; Sensors require precise placement, connection, and rigorous testing for accuracy and reliability, increasing complexity.	

## Data Availability

No data was used for the research described in the article.

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
