# Peer review of "From Sensors to Care: How Robotic Skin Is Transforming Modern Healthcare—A Mini Review"

_sensors, 2025, doi:10.3390/s25092895_

Round 1

Reviewer 1 Report

Comments and Suggestions for Authors

In recent years, robotics has made notable progress, becoming an essential component of daily life by facilitating complex tasks and enhancing human experiences. This review provides a narrative analysis of current trends in robotic skin development, specifically tailored for healthcare and rehabilitation applications, including skin types of sensor technologies, materials, challenges, and future research directions in this rapidly developing field. Some comments for improving the paper are listed as following.

  1. The introduction part simply lists a great many current literatures that provide no effective summary on the state of the art and challenging issues.
  2. The most important concern: The main contribution of the paper is not clear. What is the main difference between this specific design and other designs in the same category provided in the literature?
  3. The authors just listed some works have been conducted by researchers in the literature review part, but they did not comment on those works.
  4. Drawing of Figs 4 and 5 are very confusing, please revise it.
  5. The English of this paper should fully revised.
Comments on the Quality of English Language

The English of this paper should fully revised.

Author Response

Reviewer 1:  Comments and Suggestions for Authors

Response to Reviewer 1: Thank you very much for taking the time to review our manuscript. We truly appreciate your constructive feedback and thoughtful suggestions. Your input has been helpful in refining our work, and we have carefully considered all of your comments and suggestions. We have made revisions based on your comments and suggestions, which are highlighted in yellow. Please find our responses below.

In recent years, robotics has made notable progress, becoming an essential component of daily life by facilitating complex tasks and enhancing human experiences. This review provides a narrative analysis of current trends in robotic skin development, specifically tailored for healthcare and rehabilitation applications, including skin types of sensor technologies, materials, challenges, and future research directions in this rapidly developing field. Some comments for improving the paper are listed as following.

Comments 1: The introduction part simply lists a great many current literatures that provide no effective summary on the state of the art and challenging issues.

Response 1: On page 1, the second paragraph has been edited as follows:

However, many functions still need to be developed to mimic human skills, particularly those required to replicate human abilities, such as handling delicate or fragile objects and ensuring safe interaction with vulnerable groups like children and the elderly [1], [2]. This paper overviews some of the soft robotic skin developments and outlines the key challenges that need to be overcome to advance robotic capabilities.

We have also added a column in Table 1 to incorporate the challenges associated with sensors.

 Comments 2: The most important concern: The main contribution of the paper is not clear. What is the main difference between this specific design and other designs in the same category provided in the literature?

Response 2: On page 2, the third paragraph has been edited as follows:

This paper provides a narrative review of the current trends in robotic skin development over the past three decades, beginning in the 1990s, specifically for healthcare and rehabilitation applications, including its role in companion and healthcare robots. This review focuses on examining sensor-based robotic skin types while highlighting the key materials, technologies, challenges, and future research directions in this rapidly growing field.

 Comments 3: The authors just listed some works have been conducted by researchers in the literature review part, but they did not comment on those works.

Response 3: Tables 2 and 3 include comments on some of the works by other researchers.

Comments 4: Drawing of Figs 4 and 5 are very confusing, please revise it.

Response 4: To reduce potential confusion, we have removed Figure 4. However, we have retained and revised the corresponding discussion to clearly describe key structural configurations used in soft robotics for sensor integration. Revised sections are highlighted on pages 10-11.

We have included supplementary explanatory details for Figure 5 now as Figure 4, which is reproduced from a previously published study and serves as a representative example illustrating the application of robotic skin in advanced healthcare robotic systems. Revised sections are highlighted on pages 11-12.

Comments 5: The English of this paper should fully revised.

Response 5: Thank you for the suggestion. The manuscript has been reviewed and refined for clarity, including a check by a native English speaker.

Reviewer 2 Report

Comments and Suggestions for Authors

This manuscript focuses on the application of robotic sensors or electronic skin in modern healthcare. The authors reviewed the forms and types of sensors employed. On a thorough reading of the entire text, however, I am convinced that further revision is necessary. Several issues can be identified, as outlined below:
1. Please confirm that the tables and references used in the main text are formatted in accordance with academic standards.
2. The figures and tables presented lack typical representation and the readability factor is absent.
3. In Chapter 3, "Materials and Technologies in Healthcare and Rehabilitation Robotics," the description of materials and sensing technologies is overly vague, failing to highlight key points or specific content.
4. Chapter 4 should constitute the core section of the paper. A more comprehensive description regarding the provided images is recommended.

Author Response

Reviewer 2:  Comments and Suggestions for Authors

Response to Reviewer 2: Thank you very much for taking the time to review our manuscript. We truly appreciate your constructive feedback and thoughtful suggestions. Your input has been helpful in refining our work, and we have carefully considered all of your comments and suggestions. We have made revisions based on your comments and suggestions, which are highlighted in yellow. Please find our responses below.

This manuscript focuses on the application of robotic sensors or electronic skin in modern healthcare. The authors reviewed the forms and types of sensors employed. On a thorough reading of the entire text, however, I am convinced that further revision is necessary. Several issues can be identified, as outlined below:
Comments 1. Please confirm that the tables and references used in the main text are formatted in accordance with academic standards.

Response 1: The tables and references format have been revised accordingly.

Comments 2. The figures and tables presented lack typical representation and the readability factor is absent.

Response 2: The specific revisions required were not clear, however, we have carefully reviewed all figures and tables and have incorporated additional information to improve overall clarity and presentation.

Comments 3. In Chapter 3, "Materials and Technologies in Healthcare and Rehabilitation Robotics," the description of materials and sensing technologies is overly vague, failing to highlight key points or specific content.

Response 3: Thank you for the feedback. We have revised it to focus on the description of biocompatible and flexible materials used in robotic skin. Revised sections are highlighted on pages 9-11.

Comments 4. Chapter 4 should constitute the core section of the paper. A more comprehensive description regarding the provided images is recommended.

Response 4: Thank you for the feedback. We have revised it with a more comprehensive description. Revised sections are highlighted on page 11.

Reviewer 3 Report

Comments and Suggestions for Authors

It is a good work to review the e-skin technology in the fields of the medicial robotics and human-machine interaction. The breakthrough and key challengend are proposed, which is valuable for readers. Therefore, it is suggested to accept this work.

Author Response

Reviewer 3:  Comments and Suggestions for Authors

It is a good work to review the e-skin technology in the fields of the medicial robotics and human-machine interaction. The breakthrough and key challengend are proposed, which is valuable for readers. Therefore, it is suggested to accept this work.

Response to Reviewer 3:

Thank you very much for taking the time to review our manuscript. We sincerely appreciate your thoughtful comments and positive feedback.

Reviewer 4 Report

Comments and Suggestions for Authors

This manuscript provides a well-researched overview of current developments in the field of robotic skin, with emphasis mainly on its use in healthcare (rehabilitation). The authors describe sensor technologies, materials, application features and development prospects in detail. The work has a clear structure, contains a sufficient number of relevant references and clear illustrations, and may be useful to some extent for researchers in the field of robotics and biomedical engineering. To improve the quality of the manuscript, I recommend that you pay attention to some points.

  1. To better express the scientific novelty of the presented review of modern technologies and applications of robotic skin, I recommend providing a detailed classification of sensors, materials and technologies that can be divided according to appropriate parameters (for example, sensitivity, flexibility, biocompatibility, energy consumption and other parameters) and will be based on a critical analysis of existing solutions. We can add a comparison of the effectiveness of various technologies based on experimental data from the literature.
  2.  It would not be bad to add comparative graphs of various types of "robotic skin" (optional at the discretion of the authors).
  3. To increase clarity and informativeness, specific numerical values of parameters (measurement ranges, response time) could be added to Tables No. 2, No. 3, No. 5
  4. The conclusions section needs to be expanded and reinforced with its own specific solutions (examples) or vision, for example, based on materials or something else based on the analysis.

Author Response

Reviewer 4:  Comments and Suggestions for Authors

Response to Reviewer 4: Thank you very much for reviewing our manuscript. We truly appreciate the time and effort you invested in evaluating our work and offering valuable feedback throughout the review process. We have revised the manuscript in response to your comments and suggestions, with the changes highlighted in yellow. Please find our responses below.

This manuscript provides a well researched overview of current developments in the field of robotic skin, with emphasis mainly on its use in healthcare (rehabilitation). The authors describe sensor technologies, materials, application features and development prospects in detail. The work has a clear structure, contains a sufficient number of relevant references and clear illustrations, and may be useful to some extent for researchers in the field of robotics and biomedical engineering. To improve the quality of the manuscript, I recommend that you pay attention to some points.

Comments 1. To better express the scientific novelty of the presented review of modern technologies and applications of robotic skin, I recommend providing a detailed classification of sensors, materials and technologies that can be divided according to appropriate parameters (for example, sensitivity, flexibility, biocompatibility, energy consumption and other parameters) and will be based on a critical analysis of existing solutions. We can add a comparison of the effectiveness of various technologies based on experimental data from the literature.

Response 1: We have added additional parameters as requested for the tables. Also, in Section 5-Table 5 has provided a detailed discussion of the challenges related to power consumption, biocompatibility, and other relevant aspects.

Comments 2. It would not be bad to add comparative graphs of various types of "robotic skin" (optional at the discretion of the authors).

Response 2: We have added a paragraph discussing the various types of tactile and pressure robotic skin after the comparative analysis presented in Table 2.

Comments 3. To increase clarity and informativeness, specific numerical values of parameters (measurement ranges, response time) could be added to Tables No. 2, No. 3, No. 5

Response 3: We have added additional parameters as requested for the tables.

Comments 4. The conclusions section needs to be expanded and reinforced with its own specific solutions (examples) or vision, for example, based on materials or something else based on the analysis.

Response 4: Thank you for your feedback. We have added a paragraph discussing the contributions in the conclusion. Revised sections are highlighted on page 22.

Reviewer 5 Report

Comments and Suggestions for Authors

The article is a study on robotic skin and its application in modern healthcare. The article is well organized and has an extensive list of references (more than 180 sources), and describes a multidisciplinary approach that combines engineering and medical perspectives.
However, I have some comments for the authors:
1) What criteria were used to select sources? The review does not specify any time frame for the included studies.
2) In some sections of the article, the technical aspects of robotic skin are described in a rather general way; it would be nice to add more detailed explanations of the methods under consideration and examples from the mentioned articles. Some sections are overloaded with text and general phrases.
3) Section 6.2 states that AI improves sensory processing, but there are no examples of algorithms or architectures (e.g. CNN for tactile recognition).

For publication in the MDPI, it is necessary to add a more detailed description of the developments being studied to this work in order to enhance its practical value for engineers and physicians.

Author Response

Reviewer 5:  Comments and Suggestions for Authors

Response to Reviewer 5: Thank you very much for taking the time to review our manuscript. We truly appreciate your constructive feedback and thoughtful suggestions. Your input has been helpful in refining our work, and we have carefully considered all of your comments and suggestions. We have revised the manuscript in response to your comments and suggestions, with the changes highlighted in yellow. Please find our responses below.

The article is a study on robotic skin and its application in modern healthcare. The article is well organized and has an extensive list of references (more than 180 sources), and describes a multidisciplinary approach that combines engineering and medical perspectives. However, I have some comments for the authors:
Comments 1: What criteria were used to select sources? The review does not specify any time frame for the included studies.

Response 1: On page 2, the third paragraph has been edited as follows:

This paper provides a narrative review of the current trends in robotic skin development over the past three decades, beginning in the 1990s, specifically for healthcare and rehabilitation applications, including its role in companion and healthcare robots. This review focuses on examining sensor-based robotic skin types while highlighting the key materials, technologies, challenges, and future research directions in this rapidly growing field.

Comments 2: In some sections of the article, the technical aspects of robotic skin are described in a rather general way; it would be nice to add more detailed explanations of the methods under consideration and examples from the mentioned articles. Some sections are overloaded with text and general phrases.

Response 2: We appreciate your feedback and have made the necessary revisions. We have revised the manuscript highlighted in yellow.

Comments 3: Section 6.2 states that AI improves sensory processing, but there are no examples of algorithms or architectures (e.g. CNN for tactile recognition).
Response 3: Revised sections are highlighted on pages 20-21. Based on your suggestion, we have incorporated CNNs for healthcare and rehabilitation applications.

For publication in the MDPI, it is necessary to add a more detailed description of the developments being studied to this work in order to enhance its practical value for engineers and physicians.
We appreciate your feedback and have made the necessary revisions highlighted on page 22.

Round 2

Reviewer 4 Report

Comments and Suggestions for Authors

The authors were attentive to the comments and were able to finalize the manuscript taking them into account. Basically, my wishes were taken into account and thanks to the changes made, the material became more informative, logically coherent and better reflects the results of the study.

Author Response

Thank you very much for taking the time to review our manuscript. We sincerely appreciate your thoughtful comments and positive feedback.

Reviewer 5 Report

Comments and Suggestions for Authors

The paper looks much better. The authors have made several changes: updated the list of references, adding new relevant sources, clarified the technical characteristics of some sensors, and expanded the description of the research methodologies. Also, grammatical errors were corrected, and scientific terminology was improved, which made the text more consistent.
My only addition can only be:
Can specific examples of methods and technologies be described in more detail, as recommended?
Overall, the article covers the main topic well, but I lack a technical description of the technologies.

Author Response

We sincerely thank you for taking the time to review our manuscript and for your valuable comments and feedback. As this is a mini-review, we have intentionally focused on selected technologies. While the level of detail expected for this format may vary, we believe the current version aligns well with the intended scope of a mini-review. Additionally, as this mini-review is also limited to specific applications, the scope in this case accordingly shaped the selection of technologies discussed.